

# In-depth study of pyroptosis-related genes and immune infiltration in colon cancer

Bingbing Shang[1,2], Haiyan Qiao[1], Liang Wang[1] and Jingyu Wang[1]

[1] Laboratory Animal Center, Dalian Medical University, Dalian, China
[2] Emergency Department, The Second Hospital of Dalian Medical University, Dalian, China

## ABSTRACT

**Background**. Pyroptosis is a form of regulated necrosis that occurs in many cell and tissue types and plays a critical role in tumor progression. The diagnostic value of pyroptosis-related genes (PRGs) in colon cancer has been widely investigated. In the present study, we explored the relationship between PRG expression and colon cancer.

**Methods**. We retrieved genomic and clinical data pertaining to The Cancer Genome Atlas-Colon Adenocarcinoma from the UCSC Xena database, along with the corresponding genome annotation information from the GENCODE data portal. Utilising these data and a list of 33 pyrogenic genes, we performed principal component analysis and unsupervised clustering analysis to assess the pyroptosis subtypes. We analysed the differential expression between these subtypes to obtain PRGs, ultimately selecting 10 PRGs. We conducted Gene Ontology, Kyoto Encyclopedia of Genes and Genomes, gene set variation analysis, protein–protein interaction, and immune infiltration analyses of these PRGs. We validated the expression of TNNC1 *via* immunohistochemistry (IHC) and real-time quantitative PCR.

**Results**. After rigorous screening, excluding patients with incomplete survival data and unmatched transcriptomes, we refined our study cohort to 431 patients. We performed differential mRNA analysis and identified 445 PRGs, 10 of which were selected as hub genes. These genes were associated with various immune cell types. Specifically, *TNNC1* expression was positively associated with immature dendritic cells and NK CD56[+] cells. IHC staining indicated higher TNNC1 expression levels in tumor samples. Notably, TNNC1 expression levels were high in all the colon cancer cell lines, particularly in SW480 cells.

**Conclusion**. In this study, we explored the characteristics of PRGs in colon cancer and identified novel biological targets for early individualised treatment and accurate diagnosis of colon cancer, thus contributing to the advancement of clinical oncology.

Corresponding authors
Liang Wang, wangliang-dy@dmu.edu.cn
Jingyu Wang, wangjingyus@163.com

## INTRODUCTION

Colon cancer (CC) is the third most common cancer worldwide and the leading cause of cancer-related deaths. In 2024, a total of 53,010 patients with colorectal cancer died in the United States, including 28,700 men and 24,310 women (*Siegel, Giaquinto & Jemal, 2024*). Therefore, CC poses a significant threat to human health. Recently, many risk

factors for CC have been identified, including increased intake of hypercaloric foods, a sedentary lifestyle, and smoking. The pathological mechanisms (*Lamprecht et al., 2017*) of malignant tumours are highly complex. Numerous studies (*Chao et al., 2023*; *Christensen et al., 2019*; *Grinat et al., 2020*) have indicated that when intestinal stem cells are located at the base of each crypt in colon cancer, they can mutate and transform into cancer stem cells, potentially inducing CC.

Chemoradiotherapy is the primary treatment for CC, while other therapies include gene-targeted approaches and cancer immunotherapy (*Chalabi et al., 2024*; *Ghiringhelli & Thibaudin, 2023*). However, in advanced stages, when CC metastasises and progresses, it becomes life-threatening and no longer amenable to surgical removal. At this stage, clinical treatment options become limited, and the survival rate of patients receiving immunotherapy is only 12% (*Kalyan et al., 2018*). Moreover, given the high inter- and intra-tumor heterogeneity of CC (*Zhang et al., 2020*), effective immunotherapy requires suitable biomarkers. Hence, there is an urgent need to identify novel biomarkers to develop effective cancer immunotherapies and improve the prognosis of patients with cancer. Targeted sequencing of cancer-related genes enables precision oncology to offer personalised cancer treatment options. Currently, CC treatment is in an era of individualised therapy, relying heavily on the identification of biomarkers. Notably, combining a patient's molecular subtype with other biomarkers could offer 'precision medicine' for patients with CC, enabling individualised evaluations and tailored medication selection. This approach is crucial for improving treatment effectiveness and reducing the incidence of side effects, thereby making it a key focus of current cancer research.

Pyroptosis, a type of cell death, plays a crucial role in the pathological processes of several diseases, including various tumours, sepsis, and acute coronary syndrome (*Guo et al., 2020*; *Tan et al., 2021*; *Wei et al., 2022*). Recently, pyroptosis-related genes (PRGs) have attracted global attention for their potential to predict the prognosis of patients with cancer (*Deng et al., 2022*; *Shao et al., 2021*; *Ye, Dai & Qi, 2021*), revealing that inflammasomes recognise pathogen-associated molecular patterns. Activated inflammasomes promote inflammation (*Tsuchiya et al., 2019*) and exhibit unique morphological features under transmission electron microscopy, such as chromatin condensation, cell swelling, pore formation on cell membranes, and bubbling of plasma membranes (*Gao et al., 2022*). Moreover, pyroptosis relies on the gasdermin family of proteins, which can form pores in the plasma membrane and mediate cell death. Specifically, gasdermin D (GSDMD) was identified as the primary executor of pyroptosis (*Burdette et al., 2021*). In the canonical pathway, when caspase-1 is activated by inflammasomes, it specifically cleaves the N-terminal domain of GSDMD (GSDMD-N) to form transmembrane pores and release inflammatory factors, such as interleukin (IL)-1$\beta$ and IL-18. Ultimately, this process results in pyroptosis and cell death.

In the present study, we aimed to identify novel biomarkers that could serve as therapeutic targets for CC by comprehensively exploring the relationship between PRG expression and colon cancer. Using bioinformatic analysis, we identified 10 PRGs and differentially expressed genes (DEGs) in various pyroptosis subtypes. Subsequently, we analysed the functions of the DEGs. Furthermore, we constructed a protein–protein interaction (PPI) network to explore the interactions among the 10 PRGs and compared
their expression differences between tumor and normal groups, as well as between pyroptosis subtypes A and B. Among the identified PRGs, *TNNC1* was specifically expressed in colon cancer samples and cells, promoting further exploration of its role in CC. This study highlights the role of *TNNC1* as a potential therapeutic target for CC.

## MATERIALS & METHODS

### Data acquisition and pre-processed

Expression profiling data for CC (The Cancer Genome Atlas-Colon Adenocarcinoma (TCGA-COAD)) were downloaded from the UCSC Xena dataset. Normalised counts were converted to transcripts per million (TPM) values by selecting 'Count' as the data type. In total, 521 human colon tumor transcriptomic datasets from TCGA, comprising 480 tumours and 41 normal samples, were included in the analysis. Patient data included age, sex, and TNM stage, among others. Subsequently, the transcriptome data were matched with patient ID extracted from the clinical information. After excluding incomplete survival information and unmatched transcriptome samples, data from 431 patients were included. Genome annotation data were obtained from GENCODE (https://www.gencodegenes.org/), data types were converted based on Ensembl ID, and relevant coding genes were extracted.

### Cluster analysis

Based on the literature, a total of 33 pyroptosis genes were considered pyroptosis gene sets (*Fu & Song, 2021*; *Man & Kanneganti, 2015*). Unsupervised clustering does not rely on predefined labels, allowing the revelation of unknown patterns and subgroups, thereby providing novel biological hypotheses and perspectives. Therefore, to better identify potential biological subgroups and uncover the internal structure of data, unsupervised cluster analysis was performed based on these genes and the expression data from TCGA-COAD using the 'ConsensusClusterPlus' R package. The consensus clustering algorithm was also employed to determine the optimal number of clusters by analysing 100 iterations, thus ensuring classification stability. Principal component analysis (PCA) is a multivariate statistical analysis tool for unsupervised learning. The proportion of ambiguous clustering (PAC) is an important index for assessing the clarity of group division, where a lower PAC score indicates a clearer classification result and a more distinct sample allocation. Therefore, in this study, the consensus clustering method was combined with PAC to determine the optimum number of clusters (k). Based on the consensus clustering results, the samples were divided into subgroups A and B. Subsequently, the gene expression of PRGs was compared between these two subgroups.

### Differential analysis and PPI analysis

To identify PRGs, the R package DESeq2 (*Ritchie et al., 2015*) was used to identify DEGs across different pyroptosis subtypes. This differential expression was displayed in a volcano plot drawn using the ggplot2 package, and a heatmap was constructed using the pheatmap package. The selection conditions were |log2(fold change) |>1.0 and $p < 0.05$. To clarify their interactions, PPI analyses of the above mRNAs were performed using STRING (*Szklarczyk et al., 2019*) version 11, which is a biological database widely used for PPI analysis.

## Functional enrichment analysis and gene set variation analysis

As a common method for large-scale functional enrichment research, Gene Ontology (GO) enrichment analyses (*Ashburner et al., 2000*) include biological processes, molecular function, and cellular components. Kyoto Encyclopedia of Genes and Genomes (KEGG) (*Ogata et al., 1999*) is an important gene database containing data on genomes, pathways, and drugs. Therefore, GO and KEGG analyses were performed on DEGs using the clusterProfiler package (*Yu et al., 2012*) in R, with $p < 0.05$ as the significance judgement criterion. Based on the 'c2.cp.kegg. v7.0. symbols' gene sets, a gene set variation analysis was conducted using gene set variation analysis (GSVA) (*Hänzelmann, Castelo & Guinney, 2013*). Single-set gene set enrichment analysis (ssGSEA) was used to calculate the score of KEGG pathways according to the gene expression matrix for each sample separately, and the results were displayed as a heatmap.

## Differential gene expression analysis

The chromosome location information and TPM expression values of *HES4*, *TNNC1*, *RNF208*, *ADAM12*, *DRD4*, *POLR2L*, *RHOD*, *WDR24*, *ZNF771*, and *PRMT1* were extracted from TCGA-COAD transcriptome data. The Rcircos package was used to generate a chromosome ring map to illustrate the chromosome distribution of the 10 genes, and the Wilcoxon rank-sum test was used to compare the expression differences of these genes in the tumor and normal groups and subtypes A and B. In addition, the 'circlize' package was used to generate correlation circles among 10 genes, with gene correlations assessed using Spearman's correlation test.

## Immune infiltration analysis

Deconvolution analysis was performed on the transcriptome expression matrix using CIBERSORT to estimate the composition and abundance of immune cells in mixed cells. Gene expression matrix data were uploaded to CIBERSORT, and the samples were filtered based on $p < 0.05$ to construct an immune infiltration matrix. The distribution of the 22 immune cells infiltrating each sample was displayed in a histogram using the ggplot2 package in R.

## Immunohistochemistry

This study evaluated the expression of TNNC1 in CC tissue samples using immunohisto-chemistry (IHC). CC tissue samples were obtained from volunteers who underwent CC surgery at The Second Hospital of Dalian Medical University (Dalian, Liaoning Province, China) between January 2020 and December 2022. All participants were informed in writing about the purpose of the study, the guarantee of anonymity, and the methods of data collection and storage. Each participant signed an informed consent form. This study was approved by the ethics committee of our hospital (Licence number: 2022-036).

Paraffin sections of human colonic tissues and CC tissues were dewaxed and rehydrated according to standard procedures. Antigen retrieval was then performed in EDTA buffers. The sections were pretreated with 3% hydrogen peroxide for 10 min and rinsed three times in phosphate-buffered saline (PBS), with 3 min per rinse. Finally, the sections were incubated overnight with anti-TNNC1 antibody (1:400, 21652-1-AP; Proteintech,

Rosemount, IL, USA) at 4 °C. The next day, the protocol of a universal two-step test kit (PV-9000, Beijing Zhongshan Jinqiao Biological) was followed for the subsequent steps. Briefly, after washing to remove the primary antibody, a reaction enhancer was added to the sections, which were then incubated at room temperature for 20 min and rinsed with PBS three times (3 min, each). Subsequently, the sections were incubated with a secondary antibody (enzyme-labelled goat anti-mouse/rabbit IgG polymer) for 20 min, and colour development was achieved using nickel-diaminobenzidine (DAB). The results were assessed based on the IHC score; total IHC scores were calculated by multiplying the 'staining intensity score' (0: negative, 1: weak, 2: moderate, and 3: strong) by the 'positive rate' (0–300%) (*Li et al., 2020*). For TNNC1 expression level analysis, an expression level ≤125% was classified as the low-level group, while a score >125% was classified as the high-level group.

## Cell culture and validation of the expression level of *TNNC1*

Various CC cell lines were culture, and the tumor cells with the highest *TNNC1* expression were selected to construct stably transfected cell lines. All cells were cultured in a 37 °C (5% carbon dioxide) incubator. CCD841CoN (Meisen, Zhejiang, China), SW480, and SW620 colon cancer cells (Procell, Wuhan, China) were maintained in DMEM supplemented with 10% fetal bovine serum (FBS) and 1% penicillin-streptomycin (PS). HT29 and HCT 116 cells were purchased from Procell (Wuhan, China) and grown in McCoy's 5A medium supplemented with 10% FBS and 1% PS, respectively. The expression levels of *TNNC1* mRNA were compared between normal human colon epithelial cell lines and human CC cell lines using real-time quantitative PCR. TransZol Up (ET111-01-V2; TransGen Biotech, Beijing, China) was used to extract total RNA from the cell lines (normal colon epithelial cells and four colorectal cancer cell lines). The A260/A280 ratio of the extracted RNA ranged from 1.8 to 2.0. Total RNA concentration was measured using the Cytology 3 Imaging Reader, and an appropriate amount of RNase-free water was added to dilute the RNA to 1 µg/µL. The PerfectStart® Uni RT & qPCR kit (AUQ-01; TransGen Biotech) was used to synthesise cDNA strands by adding 1 µL of total RNA to the reaction mixture. qPCR was then conducted to detect the differential *TNNC1* expression in cells using the Applied Biosystems Step One system (Applied Biosystems, Waltham, MA. USA). The relative expression of *TNNC1* was determined using the $2^{-\Delta\Delta CT}$ method. All consumables, including gun tips, centrifuge tubes, and organic solvents used throughout the experimental process, were free of RNase enzyme contamination.

The following primer sequences were used: *TNNC1*, forward primer: 5'-CTACAAGGCTGCGGTAGAGC-3', and reverse primer: 5'-CCCAGCACGAAGATGTCGAA-3'; for *GAPDH*, forward primer: 5'-GACAGTCAGCCGCATCTTCT-3', and reverse primer: 5'-GCGCCCAATACGACCAAATC-3'.

## Statistical analysis

All bioinformatics analyses were carried out with R software version 4.0.2. Statistical analyses for IHC and qRT-PCR were conducted using GraphPad Prism 8. The significance of the variables in the two groups was assessed using the Chi-square test and F-test.

Non-normally distributed variables were compared using the Mann–Whitney U test (Wilcoxon rank-sum test), while the independent Student's $t$-test was used to assess the statistical significance of normally distributed variables.

## RESULTS

### Identification of pyroptosis subtypes

We conducted an unsupervised clustering analysis to identify two stable subgroups (Fig. 1). We observed consensus clustering from $k = 2$ to $k = 4$ in the sample clustering heatmap and tracing curves (Figs. 1A and 1B). When the unsupervised K-mean was 2, the heatmap indicated clear sample clustering, with the highest e discrimination between groups. Moreover, the consistency matrix demonstrated the strongest consensus between the groups, with no significant cross-group confounding. In addition, PCA analysis revealed that when $k = 2$, groups A and B were well separated in principal component space, indicating distinct gene expression signatures (Fig. 1C). The PAC score was significantly lower at $k = 2$ than at other k-means (such as $k = 3$ and $k = 4$), further supporting the validity of selecting $k = 2$. We then compared the expression of PRGs between the different subgroups and observed significant differences between subgroups A and B. Patients in the two subgroups exhibited different pyroptosis states (Fig. 1D).

### Differential genes and interaction network of differential pyroptosis subgroups

To assess the transcriptomic differences among the multiple regulatory modes of pyroptosis, we performed a differential analysis for each mRNA between the different subgroups, revealing 445 pyroptosis-associated DEGs (Figs. 2A and 2B). Subsequently, we constructed a PPI network for these DEGs, setting the minimum required interaction score to 0.7 to ensure high confidence. This filtering yielded 441 nodes (mRNA) and 204 edges (interaction relationships), with a mean node degree of 0.925. The interactions observed exceeded random distribution, indicating that these proteins, as a group, are at least partially biologically connected. Using k-means clustering, we divided the entire protein network into three clusters: clusters 1, 2, and 3 comprising 144, 115, and 182 proteins, respectively. This result suggests that these pyroptosis-related proteins may participate in pyroptosis *via* three distinct pathways (Fig. 2C).

### GO and KEGG enrichment analyses of pyroptosis-related DEGs

We then performed GO analysis and KEGG pathway enrichment analyses to analyse the functions and pathways associated with these DEGs. GO analysis (Figs. 3A–3C; Table S1) revealed that these DEGs participated in several biological processes such as developmental processes and hormone regulation. They are also associated with cellular components such as the extracellular region, extracellular space, and integral components of the plasma membrane. The DEGs were also enriched in various molecular functions, including receptor regulator activity, neuropeptide receptor activity, structural molecule activity, and G protein-coupled peptide receptor activity. The KEGG pathway analysis (Fig. 3D; Table S2) revealed that these DEGs were enriched in several pathways, including the

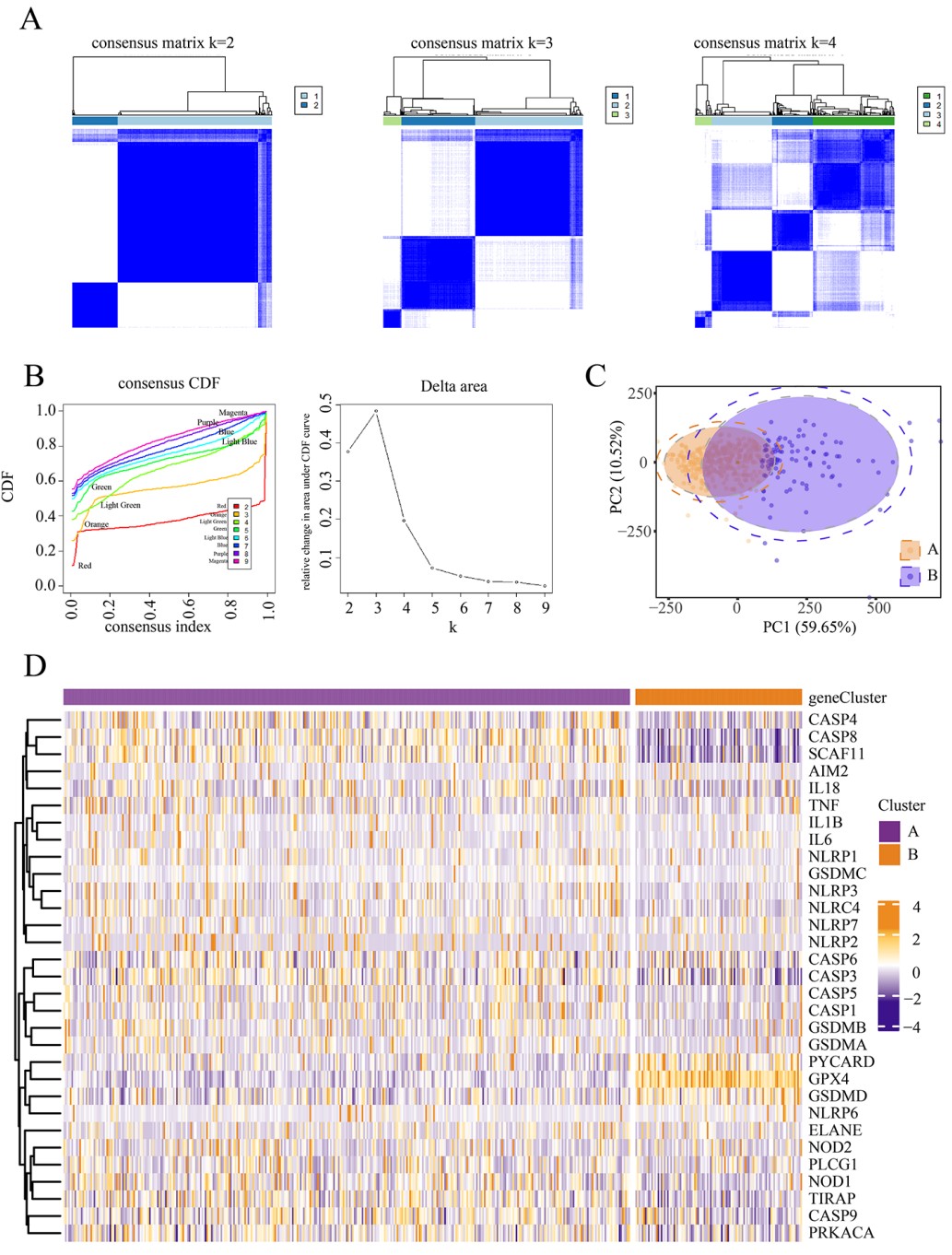

**Figure 1  Identification of pyroptosis subtype.** (A–B) Heat maps and tracing curves showed that consensus clustering grouped from $k = 2$ to $k = 4$ in sample clustering. When k was 2, the group differentiation was the highest, and the group consensus was also the best. (C) PCA analysis showed that when $k = 2$, groups A and B were well separated in principal component space, suggesting they had different gene expression signatures. (D) The expressions of pyroptosis genes are obvious differences between the two subgroups. This indicated that patients in the two subgroups represented different pyroptosis states.

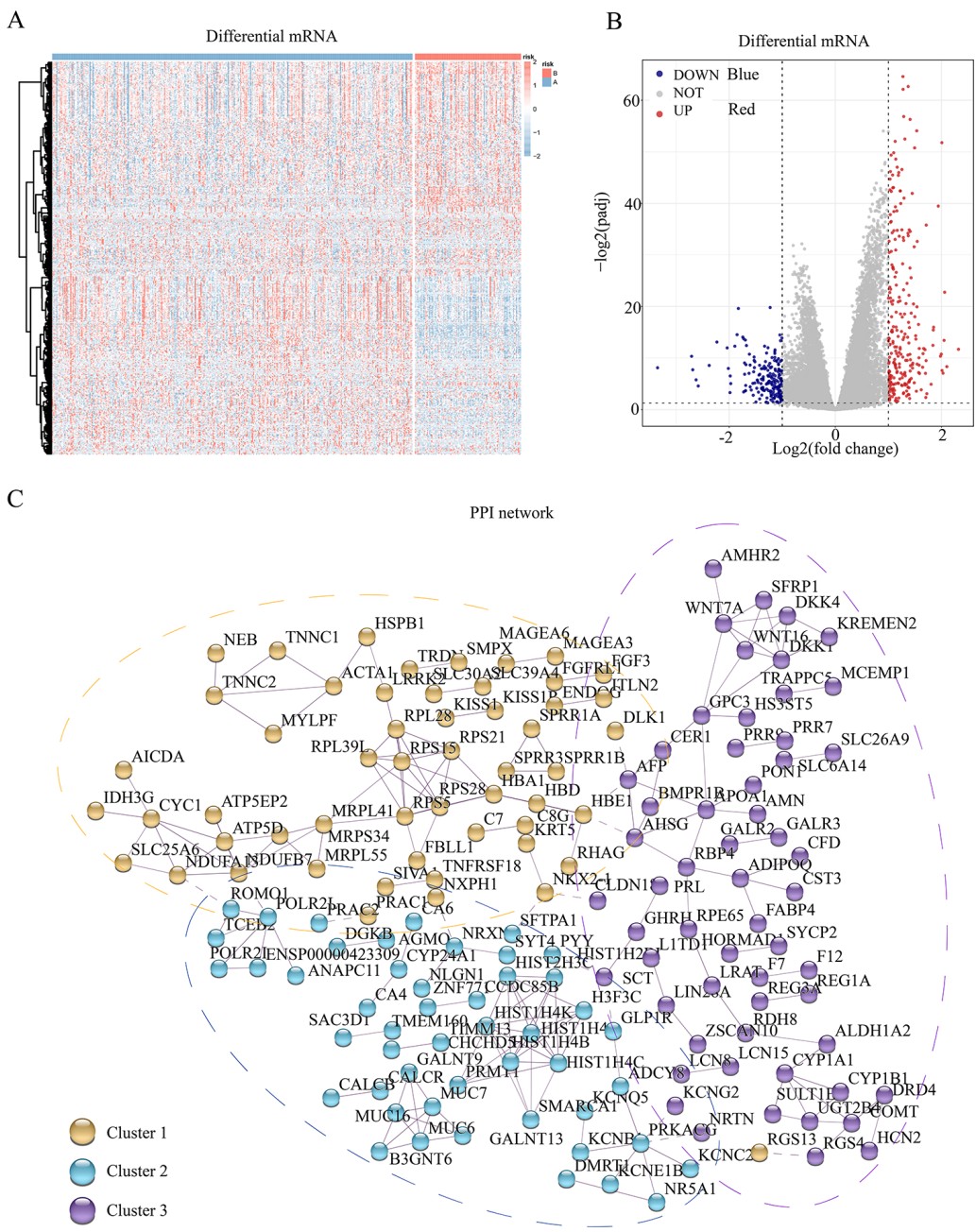

**Figure 2 Differential genes and interaction network of differential pyroptosis subgroups.** (A–B) Heat map and volcano plot of differentially expressed mRNAs in the different pyroptosis subtypes. (C) PPI showed these pyroptosis-related proteins may participate in pyroptosis *via* the three pathways.

complement and coagulation cascades, neuroactive ligand–receptor interaction, PPAR, and Wnt pathways.
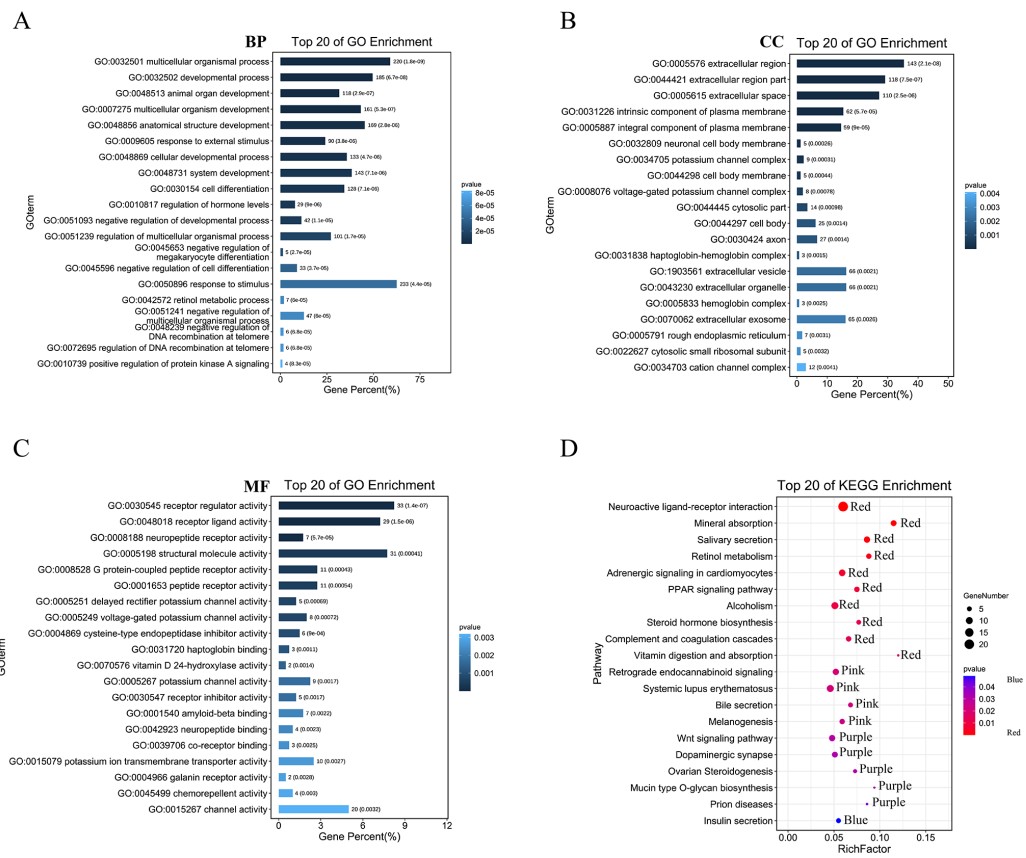

**Figure 3  GO and KEGG enrichment analysis of pyroptosis related DEGs.** (A–C) The top 20 enriched GO terms under the biological process, cell component, and molecular function categories, respectively. (D) The bubble chart of KEGG clearly showed the significant enrichment pathway of DEGs. Notes: BP, Biological Process; CC, Cellular Component; MF, Molecular Function.

## Gene set variation analysis of pyroptosis-related DEGs

We conducted a KEGG pathway analysis of the mRNA expression profiles of the different pyroptosis subtypes using GSVA. After obtaining the KEGG scores, we screened for the differential pathways between the subtypes using the limma package. GSVA analysis revealed differences in several KEGG pathways between the two subtypes; a total of 50 significantly enriched KEGG pathways are presented in the corresponding heatmaps in Fig. 4, including the transforming growth factor $\beta$ (TGF-$\beta$) signaling pathway and autophagy regulation.

## Differential gene expression analysis

We further screened candidate genes closely related to the prognosis of patients with COAD through survival and differential expression analyses. By combining these findings with existing literature on genes related to pyroptosis, we finally identified 10 genes: *HES4*, *TINNC1*, *RNF208*, *ADAM12*, *DRD4*, *POLR2L*, *RHOD*, *WDR24*, *ZNF771*, and *PRMT1*. These genes are closely related to pyroptosis processes and are significantly differentially expressed in COAD. Using a circos diagram, we illustrated the chromosomal distribution

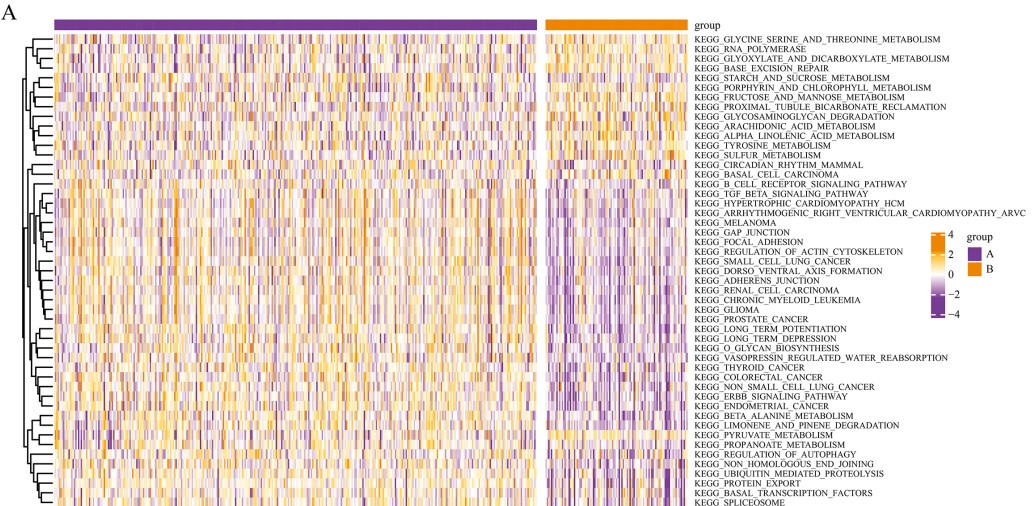

**Figure 4** **Gene set variation analysis of pyroptosis related DEGs.** Based on the GSVA of pyroptosis-related DEGs in two subgroups, differential KEGG pathway scores were plotted into a heatmap.

of these 10 mRNAs (Fig. 5A). Moreover, the expression levels of these 10 mRNAs were significantly different between normal and tumor tissues; specifically, their expression levels were elevated in tumor tissues (Fig. 5B). As shown in Fig. 5C, we analysed the correlation between their expression and various other genes using Spearman's correlation. We observed significant associations between *WDR24* and *DRD4* (rho = 0.05, $p < 0.05$) and between *WDR24* and *PRMT1* (rho = 0.66, $p < 0.05$). Therefore, we focused on the expression of these mRNAs in different pyroptosis subgroups and found that all these mRNAs were differentially expressed in various subgroups. Specifically, except for *ADAM12*, the expression of all mRNAs was elevated in subgroup B, suggesting that these mRNAs may play a role in the regulation of pyroptosis (Fig. 5D).

## Immune infiltration

We analysed the composition of immune cell infiltrates in the tumor microenvironment (TME) using the CIBERSORT algorithm. Figure 6A shows the immune infiltration profile of various cells within the TME of COAD, while Fig. 6B shows the correlation between immune cell scores. According to the immune infiltration score, we performed Spearman's correlation analysis between the expression levels of *HES4*, *TINNC* 1, *RNF208*, *ADAM12*, *DRD4*, *POLR2L*, *RHOD*, *WDR24*, *ZNF771*, and *PRMT1* and immune cells. We found that the expression of several mRNAs was correlated with immune cell infiltration (Figs. 6C–6L).

## Immunohistochemical analysis of TNNC1

In our initial screening and analysis, *TNNC1* exhibited significant differential expression in CC and was closely associated with the pyroptosis process. Additionally, *TNNC1* was identified as one of the candidate genes closely related to the prognosis of patients with CC through survival and differential expression analyses. Therefore, we conducted an IHC

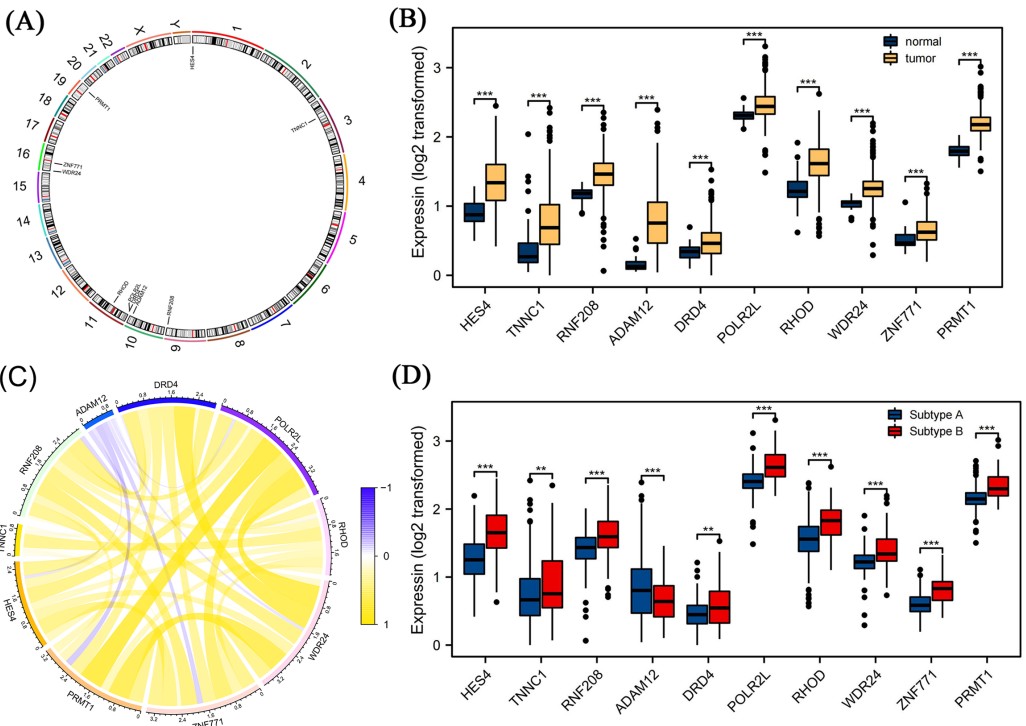

**Figure 5** **Differential expression analysis of pyroptosis related mRNAs.** (A) The distribution of ten pyroptosis-related mRNAs on chromosomes was displayed by the circos diagram. (B) Differential expression of 10 pyroptosis-related mRNA in normal tissue and tumor tissue. Compared to normal tissues, these mRNAs were highly expressed in the tumor tissues. (C) The correlation of these mRNAs. (D) Differential expression of 10 pyroptosis-related mRNA in different pyrogen subgroups. In subgroup B, apart from ADAM12, gene expression levels of the others were higher. Note: Data were retrieved from the TCGA database.

analysis to verify TNNC1 protein expression in normal colon tissues and CC tissues. In normal colon tissues, we observed that the intestinal mucosa remained intact and the gland arrangement was regular. A low level of TNNC1 expression was observed in paracancer tissues. However, unlike the normal colonic tissue, the arrangement of glands in CC tissue was disordered, and TNNC1 protein staining was positive in endothelial cells (Figs. 7A and 7B).

## Expression of TNNC1 in CC cell lines

*TNNC1* expression levels were assessed in four CC cell lines by qRT-PCR. As shown in Fig. 7C, compared to the normal colon epithelial cell line (CCD841CoN), *TNNC1* expression levels were higher in colon cancer cell lines, including HCT116, HT29, and SW620. Among the CC cell lines, the expression level of *TNNC1* was the highest in SW480.

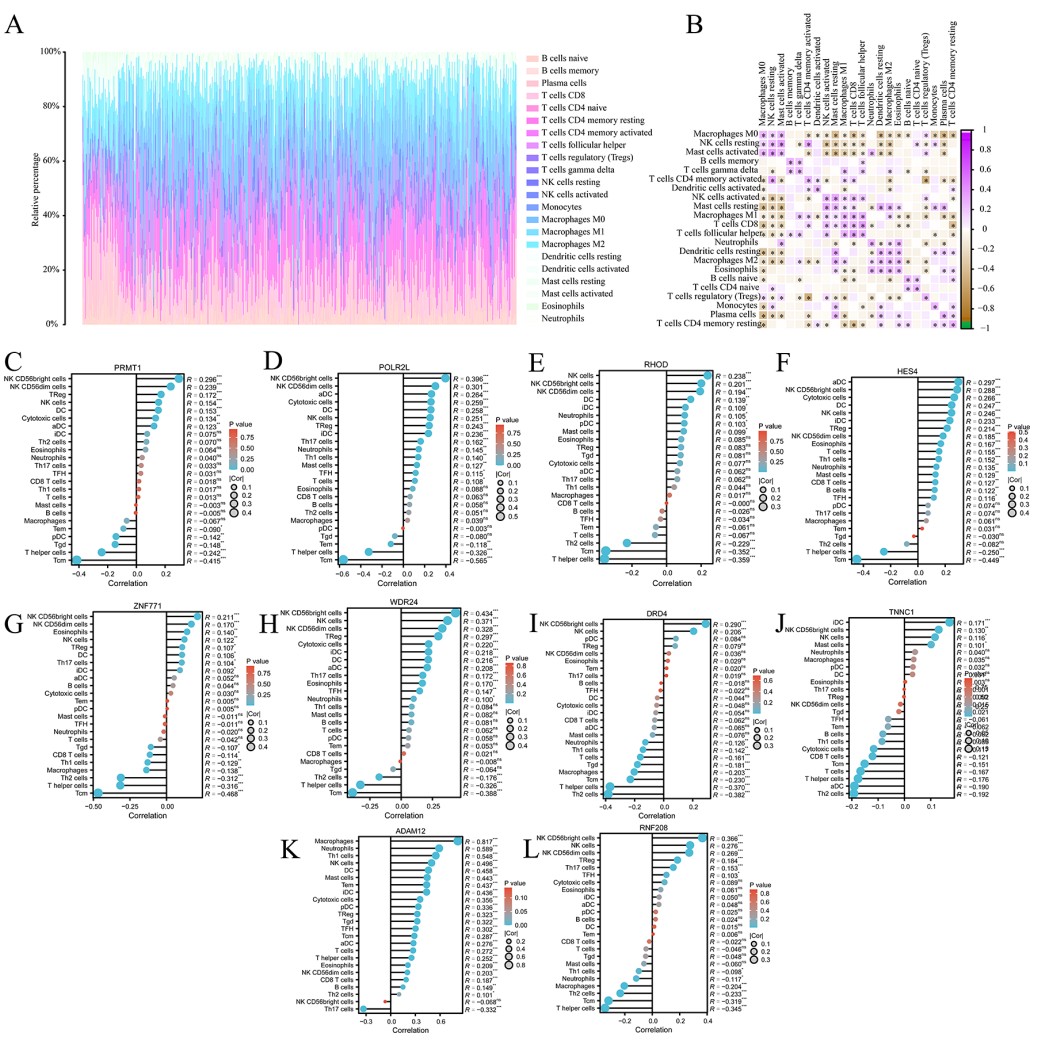

**Figure 6** Immune cell infiltration analysis of pyroptosis related mRNA. (A) The immune infiltration of panorama cells into the TME of COAD. (B) The correlation between immune cell scores. (C–L) The lollipop plots showed the correlation between 10 pyroptosis-related mRNAs and immune cell infiltrates, respectively.

# DISCUSSION

Colon cancer poses a serious threat to human health and remains a hot topic in clinical and basic research. With the rapid development of precision oncology (*Murciano-Goroff et al., 2023*), the 'precision medicine' model, which is based on individualised medical care, has gained attention. CC is a heterogeneous tumor characterised by variability in multiple gene mutations, tumor cell expression, and tumor gene expression profiles (*Challoner et al., 2024*; *Westcott et al., 2023*). In addition, there is considerable heterogeneity among different tumor cells in a patient with CC. These heterogeneities pose challenges for the diagnosis and treatment of the disease. However, targeted therapies for different patient subtypes hold the potential to improve patient survival. Pyroptosis is an inflammatory

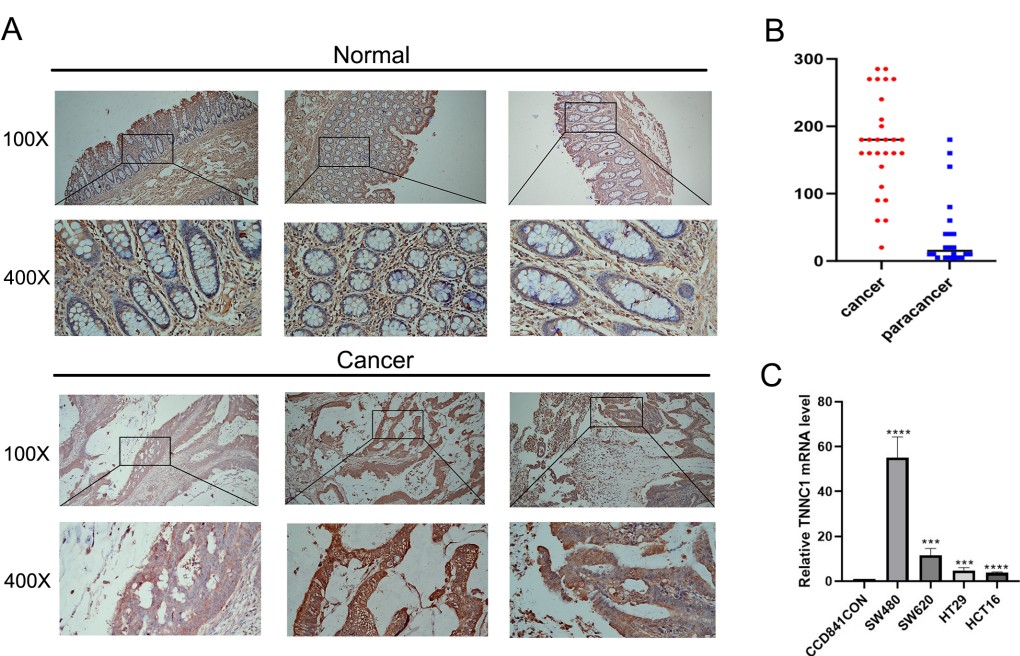

**Figure 7** **The expression of TNNC1 was verified by IHC and qRT-PCR.** (A) IHC revealed that the expression of TNNC1 in tumor tissues was significantly higher than that in normal tissues. (B) TNNC1 levels were higher in colon cancer cell lines than the normal colon epithelial cell line.

necrotic process activated by gasdermin proteins. GSDMD is a substrate for caspase-1, -4, -5, and -11. Upon activation, inflammatory caspases cleave GSDMD, leading to the formation of oligomerised GSDMD-N domains that form pores in the plasma membrane, thus disrupting the membrane and ultimately inducing pyroptosis. GSDMD serves as the executioner molecule of pyroptosis (*Tan et al., 2021*), and it is specifically cleaved by activated caspase-1 GSDMD to produce GSDMD-N terminus. Moreover, activated caspase-1 can promote the maturation of IL-1$\beta$ and IL-18 precursors. Notably, pyroptosis is associated with various multiple cancers (*Li et al., 2021*; *Liang et al., 2024*; *Xie et al., 2023*; *Zhang et al., 2021*; *Zhao & Yu, 2024*), such as gastric cancer, lung cancer, and CC. *Wang et al. (2018)* reported that GSDMD inhibits cell proliferation in gastric cancer. Recently, *Zheng et al. (2024)* demonstrated that CHMP3 could affect the development of liver cancer through caspase-1-mediated pyroptosis. These findings highlight the importance of pyroptosis in cancer, and in recent years, with the progress in mechanistic research on pyroptosis in cancer, PRGs have emerged as promising novel therapeutic targets in CC.

Through the analysis of PRGs in CC, this study revealed the key role of these genes in the diagnosis and prognosis of CC. Firstly, using established literature and databases, we screened for PRG sets that are significantly associated with pyroptosis and immune function. Based on this PRG set and TCGA-COAD expression data, we analysed PRGs in CC using bioinformatics. *Wu et al. (2023)* found different expression patterns of PRGs can not only distinguish the molecular subtypes of CC, but also predict the survival prognosis of patients. In the present study, we observed a significant difference in PRG expression

between subgroups A and B at $k = 2$, indicating that the patients in these two subgroups exhibited differential PRG expression. To assess the clinical relevance of this differential expression, we analysed the relationship between these two subgroups and tumor stage, as well as treatment response in patients with COAD. We found that some subgroups may exhibit different responses to immunotherapy. We then performed differential, enrichment, and PPI analyses to assess the functional status of the PRGs. PPI analysis indicated that these proteins could be correlated with pyroptosis *via* three different pathways. Moreover, GSVA analysis indicated differences in the TGF-$\beta$ pathway between the two subgroups. The TGF-$\beta$ pathway is a complex network involved in multiple biological regulation processes, including cell growth, differentiation, and apoptosis (*Gauthier et al., 2023*). *Sun et al. (2022)* confirmed that USF2 regulates the TGF-$\beta$/Smad3 signalling pathway to decrease pyroptosis, alleviating sepsis-associated acute kidney injury. In CC, TGF-$\beta$ activation can promote malignant progression (*Guan et al., 2024*). Notably, TGF-$\beta$ plays a critical role in regulating the stromal TME and immune resistance of CC (*Ravi et al., 2018*; *Tauriello & Batlle, 2016*). Therefore, our results are consistent with previous studies, highlighting the role of the TGF-$\beta$ pathway in CC.

In addition, we combined GO and KEGG functional enrichment analyses to identify genes associated with known immune-related pathways and strong functions, revealing *HES4*, *TINNC1*, *RNF208*, *ADAM12*, *DRD4*, *POLR2L*, *RHOD*, *WDR24*, *ZNF771*, and *PRMT1* as key genes. Notably, these genes were highly expressed in CC samples and exhibited differential expression in the two different subgroups. Pyroptosis has been shown to be associated with inflammation and tumours, and our results further reveal the functions and signalling pathways in which pyroptosis-related participate (such as complement regulatory, PPAR, and Wnt pathways).

*Ling et al. (2022)* investigated the relationship between three pyroptosis-related subtypes and tumor-infiltrating immune cells and found that these subtypes could promote individualized immunotherapy for CC. The research of *Lou et al. (2022)* releaved the pan-cancer effect of PRGs and provided a reasonable basis for the application of pyroptosis for anti-tumor immunotherapy. Likewise, we found a strong association between the abnormal expression of TNNC1 and the tumor immune microenvironment. Hence, we further explored the potential of *TNNC1* as a biomarker for the diagnosis and prognosis of CC. *TNNC1*, also known as Troponin C Type 1 (slow), was first identified as a mutant gene associated with cardiomyopathy (*Pinto et al., 2011*). As a component of the troponin complex, *TNNC1* contains a Ca$^{2+}$-binding subunit that encodes the cardiac troponin C (TnC). Different troponin subunits are expressed in non-muscle tissues and cells, including the brain, eyes, ovaries, lungs, stomach, and liver. In non-muscle cells, TNNC1 acts as a regulatory protein for cell movement (*Johnston, Chase & Pinto, 2018*). In addition, *TNNC1* is aberrantly expressed in multiple cancers, and the disruption of *TNNC1* expression is associated with tumorigenesis, particularly tumor cell invasion (*Fang et al., 2022*; *Kim et al., 2020*; *Ma et al., 2020*; *Ye et al., 2020*; *Yin et al., 2021*). For example, TNNC1 can mediate the migration of liver cancer cells through PI3K/AKT. Suppression of *TNNC1* expression can control metastasis in gastric and ovarian cancers. In lung adenocarcinoma, *TNNC1* overexpression can attenuate the invasion and metastasis of tumor cells, functioning as a

tumor suppressor gene. However, the role of *TNNC1* in the progression of CC remains unclear. In the present study, we detected TNNC1 protein and gene expression levels using IHC and qRT-PCR, respectively, and observed higher TNNC1 expression levels in CC tissues than in adjacent tissues. In addition, TNNC1 expression levels were higher in CC cells than in healthy colon epithelial cells. These findings highlight TNNC1 as a target for the treatment of CC, warranting further investigations.

The present study has some limitations. Firstly, the number of patients recruited from our hospital was small; therefore, a prognostic model could not be constructed. In the future, we will include more patients to establish a comprehensive sample information database. Secondly, although we confirmed the expression of TNNC1 in CC, we did not verify whether TNNC1 regulates the cell biological behaviour of SW480 cells. In future investigations, we will perform *TNNC1* knockdown, Transwell cell migration, scratch, and cell-counting kit 8 assay experiments to examine the inhibitory effect of siTNNC1 on tumor cell migration and proliferation. Finally, the role of TNNC1 in pyroptosis remains unclear and should be thoroughly investigated in future studies. Nevertheless, our findings may help guide future research on novel therapies for CC.

## CONCLUSIONS

In this study, we conducted an in-depth analysis of the characteristics of PRGs in patients with CC, ultimately identifying 10 hub genes. Employing unsupervised clustering analysis, we successfully segregated patients with CC into two distinct subgroups. Notably, these hub genes exhibited differential expression patterns across the two pyroptosis subgroups. Furthermore, we observed that *TNNC1*, a gene highly expressed in human CC specimens, emerged as a potential therapeutic target. Consequently, our findings offer novel biological markers that could aid in the early individualised treatment and precise diagnosis of CC, thus contributing significantly to the advancement of clinical oncology.

## ACKNOWLEDGEMENTS

The authors are deeply grateful to the patients for their cooperation. We also thank Editage for article revision and language editing.

### Funding
The authors received no funding for this work.

### Competing Interests
The authors declare there are no competing interests.

### Author Contributions
- Bingbing Shang conceived and designed the experiments, performed the experiments, prepared figures and/or tables, and approved the final draft.

- Haiyan Qiao performed the experiments, prepared figures and/or tables, and approved the final draft.
- Liang Wang conceived and designed the experiments, analyzed the data, prepared figures and/or tables, and approved the final draft.
- Jingyu Wang conceived and designed the experiments, authored or reviewed drafts of the article, and approved the final draft.

## Human Ethics

The following information was supplied relating to ethical approvals (*i.e.*, approving body and any reference numbers):

This study was approved by the ethics committee of The Second Hospital of Dalian Medical University (License number: 2022-036).

## Data Availability

The raw data is available in the Supplementary File.

## Supplemental Information

Supplemental information for this article can be found online at http://dx.doi.org/10.7717/peerj.18374#supplemental-information.

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
