# Peer review of "In-depth study of pyroptosis-related genes and immune infiltration in colon cancer"

_PeerJ, doi:10.7717/peerj.18374_

## Round 0.1 · original submission · Major Revisions

Dear Dr. Wang,

Thank you for your submission to PeerJ. In my opinion it requires some revisions as per the suggestions of the reviewers.

Reviewer 1 ·

Basic reporting

Bingbing Shang et al. studied the Multi-omic analysis of pyroptosis-related genes
correlated with immune inûltration in colon cancer. The author used the multi-omic approach to determine the 10 PRGs(pyroptosis-related genes) associated with various immune cell types. This study suggests that TNNC1 expression is positively related to immature dendritic cells and NK CD56+
cells. The study is nicely designed and well-performed.
Although, This manuscript can be accepted after minor revision.
The comments and suggestions are given below:-

Experimental design

1. In Image 7, The Author performed an immunohistochemistry assay in the normal and colon cancer tissue. Why does 400X image magnification look different in normal and colon cancer tissue? Change the images with the same magnification.
2. Use the Arrow to show TNNC1 staining in the IHC images in both normal and cancer tissue. it will help to understand the results easily.
3 There is no information provided about the antibody used for the IHC analysis. Provide complete information (antibody catalog number and dilution used for IHC ) in the material method section.
4. There is no complete information provided about the IHC assay protocol in the material method section. Add the complete IHC protocol that is used in the study (including the method used for tissue processing).

Validity of the findings

The conclusion of this study is appropriate and according to the study results.
The limitation of the study is also mentioned in the discussion section.

Reviewer 2 ·

Basic reporting

The manuscript “Multi-omic analysis of pyroptosis-related genes that are correlated with immune infiltration in colon cancer” uses bioinformatics to identify and characterize pyroptosis subtypes. The authors then focus on TNNC1 using RT-PCR and IHC analysis. The manuscript will be benefited with the following revisions.

Minor grammatical errors throughout for example use of the word “the” unnecessarily.

Use concise, descriptive, clear language throughout - much improvement needed.

Use past tense throughout when describing results.

Line 30: However, the underlying mechanisms remain unclear- do not include this if you are not addressing the mechanism.

Line 54: It has led to the death of many people every year- include specific statistic.

Line 61-62: Need to include sources for CC treatments.

Line 78: define PRGs

Make sure all figures are not blurry.

Line 87+: Do not provide a detailed description of Gasdermins if your study does not focus on them. Introduce topics relevant to your study.

You do not need to have a detailed introduction to pyroptosis in your conclusion-should be shortened to a couple sentences.

Experimental design

Line 198: You state all statistical analysis done in R and later mention using GraphPad Prism. State what statistical analysis you are using for different experiments (eg. RT-PCR, IHC).

Figure 1D: Explain how you choose the genes listed.

Figure 2: Label genes in volcano plot to show change in genes labeled in 2C.

Figure 3 legend: define BP, CC, MF

Line 249: define GSVA

Line 259: “According to the annotation information of these 10 mRNAs”- not clear what these 10 mRNAs are or why they were chosen. Need to list the gene names and explain why you choose these 10 genes.

Figure 5B and D legend: Explain the source of the data.

Line 261: Unclear what “both” is referring to.

Line 277: List names the genes that have expression correlated with immune cell infiltration.

Line 283: Provide rationale for why you are selecting TNNC1.

Validity of the findings

Figure 7A and B: Images need to be quantified-either scored by a pathologist or quantified by a computerized software.

Figure 7C: Provide statistics comparing CCD841CoN to SW620, HT29, and HCT116.

Reviewer 3 ·

Basic reporting

This manuscript aims to identify a new gene marker from a pyroptosis-related gene for colon cancer diagnosis and treatment with the bioinformatic method. Here are the points this manuscript needs to improve or address:
Minor change needed:
1: Please update the cancer statistics; 2023 and 204 data are needed. Line 54: “death of many people,” how many, world side? US number or Chinese number?
2: Please explain the abbreviations for enrichment. BP, CC, MF (understandable, but labels are needed in legend). And the method might need to be explained in the main contest to keep the flow of the story.
3: certain content needs to be edited and addressed in the way of writing. For example, the introduction and discussion talked about the pyroptosis pathway. However, none of these mentioned genes are included in the study or used as a significant marker for subtypes or clusters. Please delete these.
4: line 299 to 301: statement without citation. Please include information resources. Please check other parts of the manuscript to address citation issues.

Experimental design

1:Main logic flaw: How do you select these 10 genes? Please address these in both the main content and discussion. And from these 10 genes. How do you narrow it down from 10 genes to TNNC1? Meanwhile, please label and highlight these 10 genes in Fig2 DEGs heatmap and string cluster.
2: More information is needed in the discussion. For example:
a) Why choose the unsupervised cluster, and what is the advantage of choosing this method? Most importantly, what’s the clinical-related meaning of these two groups? Did these have any connection with the status of tumor staging? Are they correlated with the response to certain treatments, such as immune therapy?
b) During the discussion on GSVA core and data interpretation, TGF-beta signaling was mentioned. Where does this result come from? Please introduce this data in the main figures.

Validity of the findings

Please explain the data with proper details and standards. For example:
a) Fig 1 concluded that k=2 is the best grouping strategy. Why is that? What’s the proportion of ambiguous clustering (PAC) for each K value?
b) Fig 3 KEGG analysis: why choose the signaling mentioned in the context (“Complement and coagulation cascades, neuroactive ligand-receptor interaction, PPAR, and Wnt pathway”)? Is it based on gene number or p-value?
c) Fig 6: explain the correlation between selected gene and cell types. What could the correlation between cell type and the selected gene illustrate their interaction or relationship? And most importantly, a cutoff value should be established in this analysis for the correlation coefficient (0.3 or 0.5? ). Moreover, the X-axis should be the same between different genes to keep consistency.
d) Fig 7: please include more IHC stain results with representative figures (at least 6) and H-score calculations for statistics (more patient samples are needed; final data should show individual dots). Meanwhile, qRT-PCR statistics are required for other three cell lines. In addition, western lot validation was highly recommended in the analysis to support the study for both cell lines and patient tumor lysate samples.

Additional comments

The title mentioned multi-omic data analysis, while the whole analysis heavily depends on the TCGA-COAD dataset. Please include more spatial, scRNA-seq, and/or proteomic data to enhance this study.

---

## Round 0.2 · Minor Revisions

There are some minor comments remaining from Reviewer 2. In addition. one of the Section Editors (Brenda Oppert) provided the following comments which you should respond to as well:

There are quite a few recent and very similar studies not referenced here.

How is this study different from:

Wu M, Hao S, Wang X, Su S, Du S, Zhou S, Yang R, Du H. A pyroptosis-related gene signature that predicts immune infiltration and prognosis in colon cancer. Front Oncol. 2023 Jul 12;13:1173181. doi: 10.3389/fonc.2023.1173181. PMID: 37503314; PMCID: PMC10369052.

Lou, X., Li, K., Qian, B. et al. Pyroptosis correlates with tumor immunity and prognosis. Commun Biol 5, 917 (2022). https://doi.org/10.1038/s42003-022-03806-x Lei R, Li S, Yu G, Guan X, Liu H, Quan J, Jiang Z and Wang X (2021)

Deciphering the Pyroptosis-Related Prognostic Signature and Immune Cell Infiltration Characteristics of Colon Cancer. Front. Genet. 12:755384. doi: 10.3389/fgene.2021.755384 Song, W., Ren, J., Xiang, R., Kong, C., & Fu, T. (2021).

Identification of pyroptosis-related subtypes, the development of a prognosis model, and characterization of tumor microenvironment infiltration in colorectal cancer. OncoImmunology, 10(1). https://doi.org/10.1080/2162402X.2021.1987636 Ling Y, Wang Y, Cao C, Feng L, Zhang B, Li S.

Molecular subtypes identified by pyroptosis-related genes are associated with tumor microenvironment cell infiltration in colon cancer. Aging (Albany NY). 2022 Nov 16; 14:9020-9036 . https://doi.org/10.18632/aging.204379

https://doi.org/10.1155/2022/4492608

DOI:10.1155/2022/2035808 And so on. "

Please expand on how your study contributes to the diagnosis of colon cancer in light of these other studies.

Reviewer 1 ·

Basic reporting

The Manuscript can be accepted in its present form

Experimental design

The study experiment designed well and nicely performed

Validity of the findings

The conclusion of this study is appropriate and according to the study results.
The limitation of the study is also mentioned in the discussion section.

Additional comments

The Manuscript can be accepted in its present form

Reviewer 2 ·

Basic reporting

Overall, the language quality is greatly improved. Please fix minor issues below.
Line 188 fix spelling of fetal (in FBS)
Line 275 replace “mRNAs” with “genes”
Text in Figure 6 C-L is blurry and not readable- need to make non-blurry
Line 388: remove “the” before TNNC1 in “These findings highlight the TNNC1 as a target”

Experimental design

IHC in Figure 7A must be quantified.

Validity of the findings

No comment.

---

## Round 0.3 · accepted · Accept

Dear Dr. Wang,

Thanks for your sincere efforts in revising the manuscript.